# Finite Boundary Conditions Due to the Bar Presence in the Model of Chloride Penetration

**DOI:** 10.3390/ma15041426

**Published:** 2022-02-15

**Authors:** Fabiano Tavares, Carmen Andrade

**Affiliations:** 1Department of Mechanics of Continuum Media and Structures, Polytechnical High School of Cordoba, Campus Rabanales, 14031 Cordoba, Spain; ftavares@uco.es; 2International Center for Numerical Methods in Engineering (CIMNE), 28010 Madrid, Spain

**Keywords:** reinforcement corrosion, chlorides boundary conditions, service life

## Abstract

The chloride penetration is usually modelled through the application of a solution of Fick’s second law of diffusion, based on the assumption of semi-infinite boundary conditions. However, the presence of the bars, on whose surface the chlorides accumulate, makes this assumption incorrect. As the time progresses, the chlorides in the steel/concrete interface increase in concentration more than the chlorides overpassing the bar position without obstacles. This circumstance, although previously studied, has not been introduced in common practice, in spite of it supposes early reaching of the chloride threshold. The study in this paper shows a deterministic analysis of the chloride diffusion process by the finite element method (FEM) which numerically solves Fick’s second law, taking into account the accumulation of the chlorides on the bar surface. Several examples are calculated and factors between the finite/semi-infinite solutions are given. These factors depend on the cover depth and the diffusion coefficient, and with less importance, on the diameter of the bar, which make it unfeasible to propose a general trend.

## 1. Introduction

In marine or chloride bearing environments, the chloride ions diffuse through the concrete pores, and when they arrive to the reinforcement surface in a concentration named “critical chloride content”, corrosion develops [1]. The rust produced has an expansive nature occupying a higher volume than the original materials and in most of the cases originating the cracking of the concrete cover [2].

This process has generated numerous studies, either on the phenomenon of the transport of ions through the concrete pores and its modelling [2,3,4,5,6,7,8,9,10,11], as in the decrease of the diffusion coefficient with time (aging factors) [12,13] or the critical chloride concentration that induces steel depassivation [14,15,16,17]. In particular, chloride transport has attracted a huge research interest in the last decade, in authors’ opinions from the introduction of probabilistic models in the fib Model Code 2010 [18], and perhaps also due to the computing calculation progresses made from the beginning of the XXI century, around 20 years ago. The increase in research studies has been also promoted by the awareness on the impact of premature corrosion in the economic expenditures in too relatively young structures that need urgent repair [19,20]. Both availability of models and need of having structures with long service life are some of the most important reasons for the exponential increase in studies on this subject. In spite of this huge number of papers, some uncertainties remain [21]. Numerous papers are only confirming previous findings, by using other variables as concrete mix proportions, types of binders which cause slight changes in the testing conditions, or making general reviews that are interesting, but do not propose solutions [22,23,24] to the still pending open questions on the long term prediction of the chloride ingress in concrete. Neither has found accurate enough practical engineering solutions that could be applied in the daily design of structural concrete. The interest in the subject, then, will continue until such an engineering design method could be calibrated with performance in real environments.

As it is impossible to cite all the papers published in subjects related to chloride transport in concrete in the last few years, only some of them will be commented next, related to some of the open questions (general models, aging effects, surface concentration evolution, binding effects depending on the binder type, among others):-General models are reviewed and commented in [22,23,24] being the models proposed of increasing complexity, as they are more detailed. The complexity appears due to the multiple variables concerned in the material, adding its evolution in time and the modifications introduced by the interaction with the cycling and evolutionary environment. All parameters together mean that the number of basic phenomena influencing each other, and of the model input parameters and constitutive laws to be fulfilled, is very high. Thus, Ref. [22] analyses several different models mentioning their “accuracy and precision” while the results are limited in time, logically because they are not validated in the long term, which makes the “precision” a non-homogeneous concept as each model has been calibrated with its own results. The work in [23] has developed a comprehensive model, but whose input parameters are very difficult to be obtained in normal experiments and also has not been calibrated beyond short term results. In [24] is presented also a model based in thermodynamic equilibria data and surface complexation which is promising, as it seems to model well the chloride binding isotherms, but again it has not been validated in real conditions in the long term. Identifying that present models have a limited amount of variables but the process is more complex, in [25] the authors extended through “Hermite polynomials non-constant multifactorial” to a more complete model, taking into account the water/cement ratio, chloride binding, temperature, age, deformation, damage, and humidity. A similar study was made through another numerical tool [26], through a reactive–transport model.-With respect to the input parameters as is the chloride surface concentration, in some cases it is found that the more superficial zone of the concrete performs differently to the bulk. This is named as concrete “skin” with a different diffusion coefficient [2,18,27]. This superficial (skin) zone having different diffusion properties than the bulk, in [18] is named the “convection” zone, in spite of convection is not the only mechanism responsible for the effect, because also the concrete composition is different in that zone near the surface, due to the “wall effect” of the mould, which generally induces a higher amount of mortar and paste than of large aggregates in the that “skin” zone. The mechanism supporting the development of such a skin zone has been studied from different assumptions. Thus:○In [28] by introducing an empirical manner to take into account that the chloride profile may have the maximum chloride concentration in the interior of the concrete and not on its surface. This work, however, does not give a model on how this maximum evolves in time, which is a central question for long term prediction. On this aspect of the maximum of chloride concentration [29], studies the effect of rainfalls in the chloride concentration at the concrete surface, concluding that the concentration decreases with time (instead of increasing); however, other simultaneous effects are not included in the study. In [30] is considered the convection–diffusion transport in the external parts of the concrete under reverse water pressure and makes a model through FEM, while [31] studies the effect of capillary action on the chloride transport in that zone. Another perspective is taken to explain the maximum by [32,33,34] because they consider the action of the carbonation as a promoter of the decrease in the chloride binding capacity and then, on changing the shape of the chloride profile.-With respect to the bound and free chlorides, in [35] is studied several cases of chloride isotherms, resulting that a solid solution is found between the Friedel’s salt and hemicarboaluminate, together with the known fact that chlorides are adsorbed on the calcium silicate gel, pointing out that the method of measurement of the free and bound chlorides are a key aspect for the correct quantification. Authors in [36] also studied the effect of the testing condition defining that the isotherms have been studied in a diluted solution which affects the equilibrium of free/bound chlorides and also modifies the pH of the solution. Moreover, Ref. [37] studied the chloride adsorption on the C-S-H, but attributing to the electrostatic charges the binding ability, which changes with the nature of the cation. However, Ref. [38] emphasizes on the subject the effect on the adsorption of aluminium’s incorporation into C-S-H gel. The cation effect is as well addressed by [39] and they found that more chlorides were bound when the associated cation was Mg^2+^ or Ca^2+^ compared to Na^+^. Additionally, Ref. [40] addressed the effect of the cation but with the simultaneous change in pH of simulated pore solutions. In this subject, Ref. [41] investigated the effect of the temperature on Friedel’s salt decomposition.-Finally, with respect to the chloride threshold, several reviews were made collecting data of the literature [17,42,43,44,45,46]; however, due to the several influences acting simultaneously, until present, the best consists in the prediction of the chloride developing corrosion through a statistical distribution [14,16].

The most used model in all these papers, with some small differences to calculate the time taken by the chlorides to reach the critical concentration, is the solution of Fick’s second law, Equation (1) [2].
(1)$$-J(x)=\frac{\partial C(x)}{\partial t}=D_{ap}\frac{\partial^{2}C}{\partial x^{2}}$$
where:-*x* is the distance from the concrete surface (cm),-*y* is the time (s),-*J*(*x*) is the chloride flow (g/cm^2^·s),-∂*C*(*x*)/∂*t* is the variation of the chloride concentration at a precise distance from the surface (g/cm^3^),-Dap is the diffusion coefficient (cm^2^/sg),-∂*^2^C*/∂*x^2^* is the derivative or variation of the concentration with the distance (g/cm^3^).

Assuming semi-infinite boundary conditions and that at *x* = ∞ *C_x_* = 0, the solution is shown in Equation (2) resulting in the so named “error function Equation” (2):(2)$$C_{x}=C_{s}\left(1-erf\frac{x}{2\sqrt{D_{ap}t}}\right)$$
where:-*x* is the distance reached by the chloride concentration defined as *C_x_* (cm). If a “skin” superficial zone is considered, the term “*x*” is replaced by (*x* − Δ*x*) [18] being Δ*x* the thickness of that skin layer,-*C_x_* is the concentration of chlorides at a distance in a time. When *C_x_* reaches the threshold value, the steel depassivates (% by cement weight),-*C_s_* is the concrete surface concentration of chlorides in the first mm of the concrete surface (% by cement weight),-*t* is the time (s).

In all these equations, semi-infinite boundary conditions are considered which means that the distance from the source (the external solution in contact to the concrete surface) to the interior is infinite, with “semi” meaning that the diffusion is produced only towards one direction (the interior of the concrete). However, this assumption is not correct in the places were a rebar exits, because the presence of the rebar supposes an obstacle positioned relatively near the concrete surface. Instead of infinite boundary, when the chlorides arrive to the bar surface, an accumulation of chlorides is produced, being then the boundary, finite. This fact was previously identified in [47], not finding this publication a further interest in any of the literature references mentioned. The consequences of the chloride accumulation at the bar surface, nevertheless, should not be neglected, as the service life results as being shorter because the earlier achievement of the critical concentration *C_x_* at the bar surface aims into an earlier depassivation.

The calculation of such an accumulation effect cannot be exact enough by using analytical expressions. Its calculation has to be made through the use of numerical methods of calculation [48]. In the present paper is analysed, through a Finite Element Method, how many chlorides accumulate at the rebar surface in different situations, and which are the consequences in the service life calculation of not considering such finite boundaries and consequent chloride accumulation. Relation of finite/semi-infinite results are named “semi-infinite/finite conversion ratio” and they are analysed with respect to their possible generalisation in order to facilitate the direct calculation from the present analytical Equation (2).

## 2. Experimental

It is in assumed unidirectional diffusion, as indicated in Figure 1-left, where it is shown, very schematically, the penetration of chlorides from one surface side. This process can be modelled by FEM employing the Galerkin weighted residual method, and introducing the boundary as a barrier interrupting the transport of the chloride ions [48]. Such a barrier can be expressed mathematically by the Neumann boundary condition, Equation (3):(3)$$\frac{\partial c}{\partial n_{i}}=0$$
where $n_{i}$ is the normal vector on the interface rebar/concrete. For the mesh, a linear triangular element with straight sides and a node at each corner (Figure 1-right) is used. There is one degree of freedom associated to each node for a field variable chloride concentration.

For that study, an interface to work with two external programmes was developed:-GMSH: an automatic 3D finite element grid generator post-processor.-OOFEM—free finite element code from Czech Technical University, Prague, Czech Republic, that was modified with the author’s authorisation.

For studying the effect of the accumulation, several theoretical examples were numerically calculated considering the variation of the:-bar diameter,-cover depth,-value of the diffusion coefficient,-service life duration.

The conversion ratios between finite to semi-infinite boundary conditions will be calculated for the time to depassivation (F_years_) and for the critical chloride concentration (F_Cx_).

## 3. Results

First, the one dimensional (1D) results assuming the chloride concentration with respect to the cement content with bars from 8 to of 40 mm in diameter will be presented, while in the Example set number 2, the chloride concentrations will be referred to as the concrete mass with bars of only 20 mm. Finally, 2D representations will be shown to illustrate the effect of the diameter and of the corner position of the bar in the difference between the finite and the semi-infinite boundary conditions.

### 3.1. D Representations

Figure 2 shows the chloride concentration after 25 years at the rebar level of 1 cm for a D = 10^−9^ cm^2^/s. With a critical concentration of chlorides, *C_x_* = 0.4%, the time to depassivate the reinforcing bar with the classical error function solution (Equation (2)) is 8220 days (around 22 years and a half) while if the bar position is considered, the time would be of 1080 days (around 3 years). Thus, the overestimation of the service life in the case of semi-infinite boundary conditions would be of almost 20 years.

Figure 3 shows the chloride profiles for different rebar positions with the same D = 10^−9^ cm^2^/s and t = 25 years, as in Figure 2. The overestimation of the time to corrosion initiation decreases with the cover depth in such a manner that, for cover depths above 3 cm, the differences are not significant, which is explained because the D value of 10^−9^ cm^2^/s is very low.

In Figure 4 is represented the chloride concentration versus the depth of *C_x_* for a diffusion coefficient of D = 10^−8^ cm^2^/s. Now the influence of the cover depth is much more important and cover depths of 3 cm give very different critical chloride concentrations at fixed times in the finite boundary case than when not considering the presence of the rebar (the classical error function solution with semi-infinite boundaries).

Figure 5 depicts the chloride profiles with D = 10^−9^ cm^2^/s for 50 years’ service life instead of 25 years. At this age the cover depth of 3 cm, assuming the finite boundaries presented a difference to the semi-infinite ones (Equation (2)) while at 25 years, resulted in the same value. This enables one to deduce that the overestimation of Equation (2) (the classical error function) is higher as the time (service life) is longer.

The effect of the bar size is shown in Figure 6 for bars of 2 cm and 4 cm in diameter. The concrete cover selected for the example was of 3 cm and a D = 10^−8^ cm^2^/s. The results indicate a difference of around 0.1% in chloride concentration at the rebar level after 25 and 50 years of exposure.

Figure 7 shows the same, but for other bar diameters and a smaller D = 10^−9^ cm^2^/s. The differences at 25 and 50 years can be quite important for thicker bars.

### 3.2. D Representations

Following with the effect of the bar diameter, in Figure 8, the chloride concentration distribution for two different diameters (left diameter of 40 mm and right diameter of 20 mm) is shown. The critical concentration will arrive first at the thicker rebars because higher diameter represents a larger barrier for the chloride ions flux.

With respect to the corner position effect, Figure 9 shows the numerical simulation in two dimensions for the case of a C_s_ = 1% by cement mass and a D = 1 × 10^−8^ cm^2^/s. The external concentration, C_s_ is assumed to be constant in two sides of the sample, as can be seen on Figure 9b–d. This Figure also shows the distribution of chloride concentration near the rebars and how the chloride reaches each bar at different times.

The Figure 9a shows an example in which three different points, P1, P2 and P3, are specifically studied. P1 is marked in the left side of the upper figure and it is at the same depth as the bar, but in a zone where no obstacles for the chloride diffusion exit. This point P1 will show the concentration of chlorides that follows semi-infinite boundary conditions. The points P2 and P3, however, are in the surfaces of the bars (one in the corner and the other in the middle of the concrete bulk). They will notice the accumulation of chlorides due to the finite boundary conditions, that is, the importance of taking into account the presence of the rebars. The point P3 will show the phenomenon of corner effect.

The concentration of chlorides in these three selected points are shown in Figure 9b–d. Thus, the critical concentration of *C_x_* = 0.4% reaches, first, the region of point P3 (at about 8.4 years) in spite of it being located deeper inside the concrete in comparison to P2 and P1. Figure 9 also shows that for the same time, points P1 and P2 have different concentrations because of the presence of the rebar and the position of them. The critical concentration *C_x_* is going to reach, first, point P2 (at about 12.1 years) and it is going to take double the time to reach point P1 (at about 20.2 years).

Figure 10 depicts the evolution of the chloride concentration with time for the points P1, P2 and P3 of Figure 9. It shows clearly that the bar in the corner (point P3) is, each time, higher than in the positions P2 and P1. This Figure 10 shows that the difference in the estimation of the service life is about 11,8 years (for the same *C_x_*). This time represents the difference between that one when the critical concentration reaches the rebar at point P3 and the time that the *C_x_* concentration reaches P1.

Another view of the effect of chloride accumulation on the external surface of the bar is shown in Figure 11-left, which visualises that at the higher (most external) bar surface, the chloride concentration is higher than that in the zones without a bar. In the right part of the figure is shown the aspect of the chloride front when it had gone beyond the bar positions, and how, in the side-zones where there are no bars, the chloride concentration is higher than at the bar surface, as well as while the chloride concentration is deeper in the zones where there are no bars. The example is made for a surface concentration of 0.25% by the weight of concrete, a critical chloride concentration of 0.05%, a cover depth of 50 mm and a bar diameter of 20 mm. The aging factor applied was *m* = 0.4. The calculation of the aging factor is made through the equation:(4)$$D\left(t\right)=D_{0}{\left(\frac{t_{i}}{t_{0}}\right)}^{-m}$$

-*D (t)* is the diffusion coefficient when it decreases with time (cm^2^/s),-*D*_0_ is the Diffusion coefficient at the testing time *t*_0_ (cm^2^/s),-*t*_0_ is the time at testing the chloride diffusion (s),-*t_i_* is the time for the service life (s),-*m* is the aging factor (-).

**Figure 11 materials-15-01426-f011:**
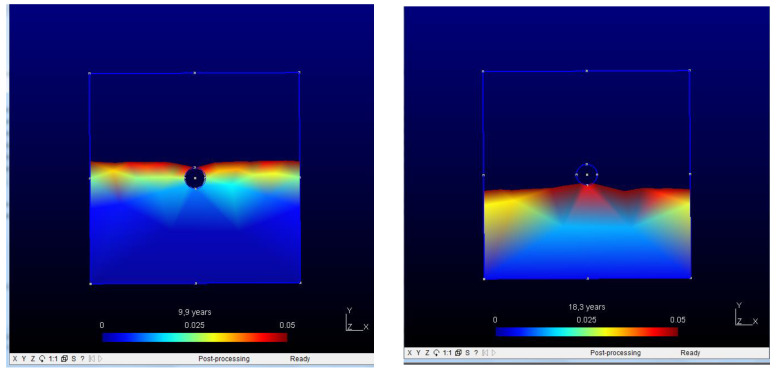
(**Left**) This part visualises the critical chloride front position when arriving at the bar. (**Right**) This part shows when the chloride front passes beyond the bar surface.

## 4. Discussion

The ratio between the finite/semi-infinite solution of Equation (1) is not constant due to the several input parameters that can be varied [21,48]. As a particular illustration, Table 1 gives the overestimation of the time to corrosion initiation (the arrival of the *C_x_*) of the classical Erf-Function equation in comparison with the results taking into account the rebar position (finite boundary). The concrete cover for those simulations is 1 cm.

In Figure 12, the ”years–ratios” (*y*-axis) of time to depassivation in the function of the cover depth (*x*-axis) for some common diffusion coefficients and a surface concentration of 0.25% by concrete mass and a critical chloride content *C_x_* = 0.05% by concrete mass, are shown. The full lines are not taking into account an aging factor (a decrease of the diffusion coefficient with time), while the dashed lines are for an aging factor m = 0.2, following Equation (3). The shortening of the time to depassivation is higher as the diffusion coefficient gets higher, while it is much higher when an aging factor is applied.

The figure indicates that as the cover depth is higher, the impact of the boundary conditions is smaller, that is, the finite/semi-infinite conversion ratio increases, and also, that the ratio is smaller as the diffusion coefficient is smaller.

With respect to the content of chlorides arriving to the external bar surface, Figure 13a shows the quantities calculated after 50 years for a diffusion coefficient of 10^−8^ cm^2^/s, considering finite or semi-infinite boundary conditions. In Figure 13b are depicted the finite/infinite ratios for the diffusion coefficient of 1 and 2.5 × 10^−8^ cm^2^/s.

## 5. Conclusions

In previous works, it has been mentioned that present models based in Fick’s second law of diffusion, with a solution considering semi-infinite boundary conditions, do not always follow the real process, and its boundary conditions for solving the differential equation in non-steady state conditions do not take into account the presence of an obstacle as the rebar is. Thus, the bar represents a barrier for the chloride penetration, and then the ions accumulate at the bar’s surface, aiming to increase the chloride concentration, with respect to when the calculation is based in the classical error function equation resulting when solved considering semi-infinite boundaries (Equation (2)). Through the use of a finite element program, the consequences in the predictions of considering finite boundary conditions for the bar position have been shown.

From the results, it can be concluded that the critical chloride threshold concentration, *C_x_*, will be reached faster at the bar surface:-if the cover is thinner,-if the diffusion coefficient is higher,-if the service life is longer,-and if the bar is thicker.

There have been calculated the ratios between the finite to the semi-infinite solution. However, the variability of the input parameters prevents the possibility of a generalisation. It is necessary to have a larger number of examples to try to identify some general trends in the ratios that could be applied in a simpler manner without the need to make numerical calculations.

## Figures and Tables

**Figure 1 materials-15-01426-f001:**
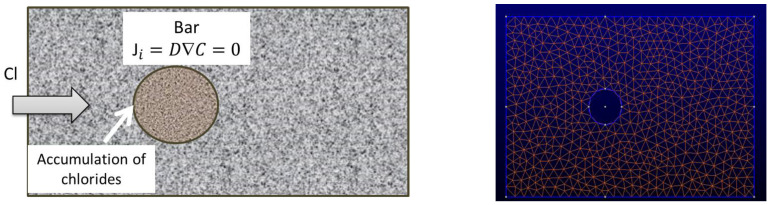
(**Left**) Rebar is acting as a barrier against the chloride diffusion where J*_c_* is the chloride flux. (**Right**) Mesh used.

**Figure 2 materials-15-01426-f002:**
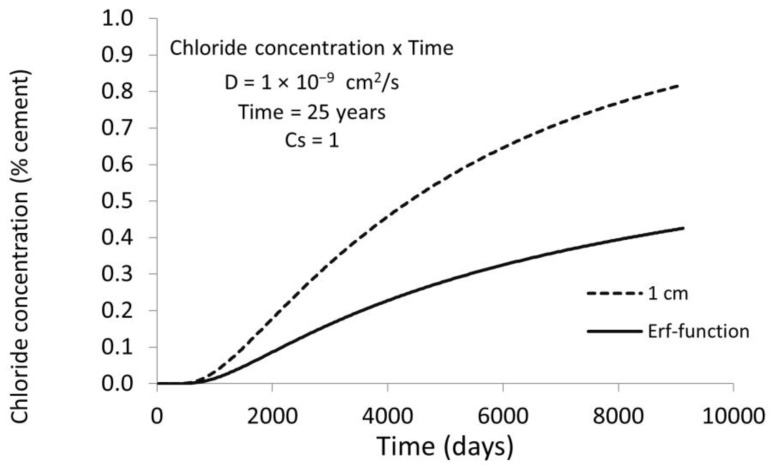
Chloride concentration (%) vs. time for a concrete cover of 1 cm: dashed line indicates the chloride concentration with time taking into account the presence of the bar (finite boundaries) and full line with semi-infinite boundaries. The concentration increases much rapidly in the bar surface than out of the bar in the concrete bulk.

**Figure 3 materials-15-01426-f003:**
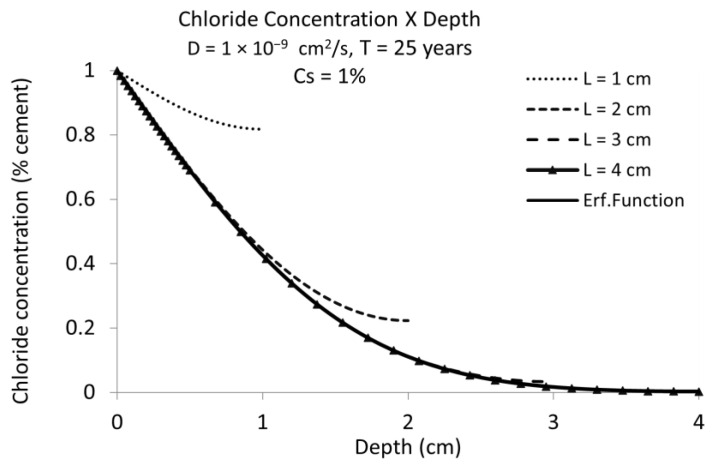
Chloride concentration vs. cover depth for chloride diffusion coefficient D = 10^−9^ cm^2^/s during 25 years. Full line shows the semi-infinite boundary results while the dashed lines give the chloride concentration for several cover depths taking into account the bar presence (finite boundary solution). The effect of the finite boundary is more important as the cover depth is smaller.

**Figure 4 materials-15-01426-f004:**
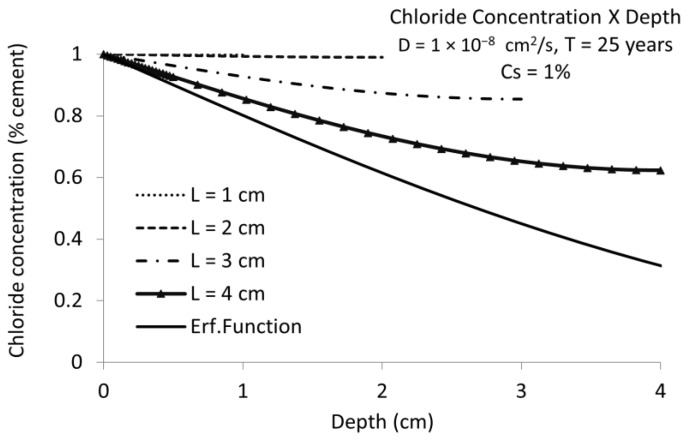
Chloride concentration vs. cover depth for chloride diffusion coefficient D = 10^−8^ cm^2^/s during 25 years. The same as Figure 3 but for higher diffusion coefficient. Full line (named “Erf. Function”) is the result in semi-infinite conditions. The rest of curves are for finite boundary conditions (considering the presence of the bar).

**Figure 5 materials-15-01426-f005:**
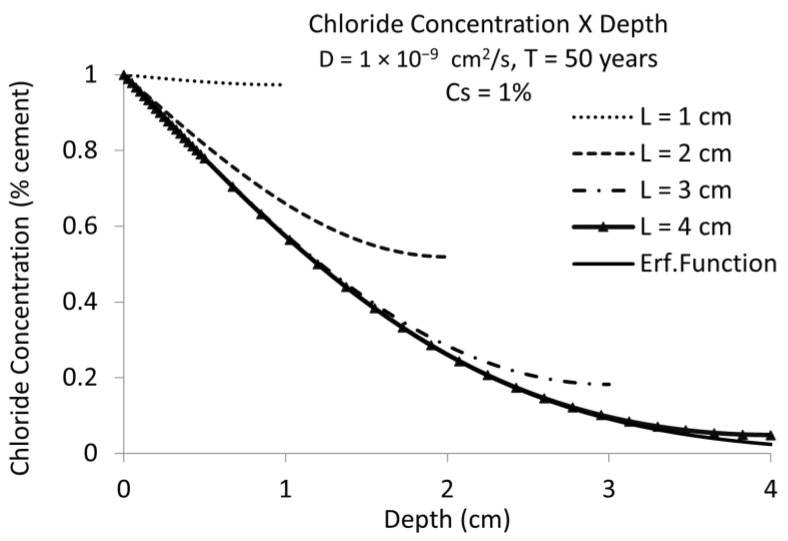
Chloride concentration versus concrete depth considering chloride diffusion coefficient D = 10^−9^ cm^2^/s during 50 years. The same representation as Figure 4 but for another diffusion coefficient and service life time. Full line named “Erf. Function” in semi-infinite conditions results very similar to that with triangles for the bar presence at 4 cm of cover depth.

**Figure 6 materials-15-01426-f006:**
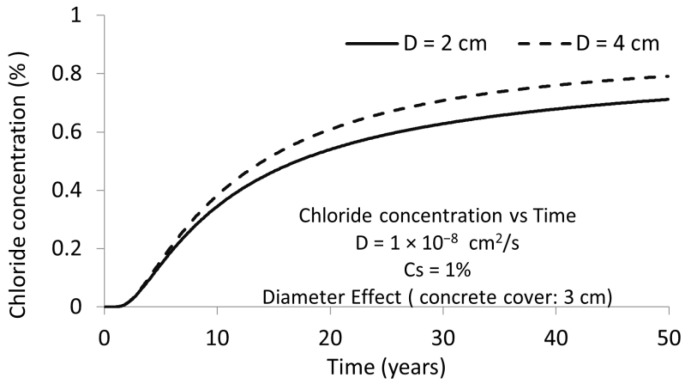
Chloride concentration versus time for two diameters at a same concrete cover, both with finite boundary conditions. The chloride increases more rapidly in the larger diameter.

**Figure 7 materials-15-01426-f007:**
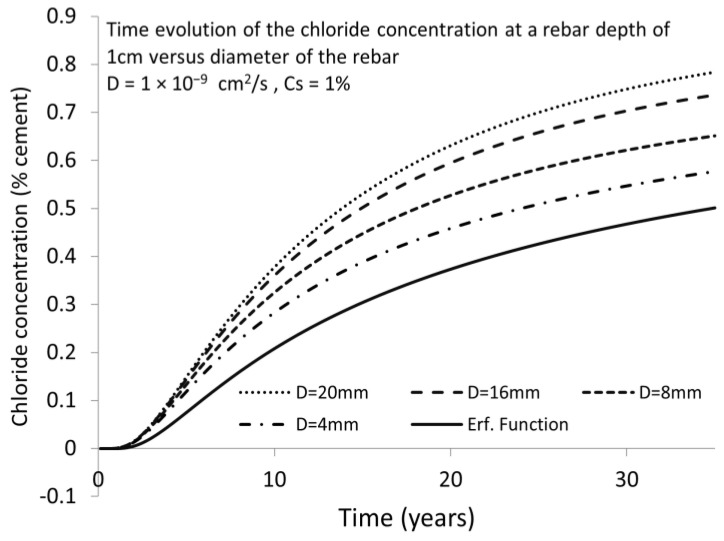
Chloride concentration variation in finite boundary conditions for different rebar diameter and a lower diffusion coefficient than in Figure 6, with the same concrete cover. Full line named “Erf. Function” is showing the results for semi-infinite conditions.

**Figure 8 materials-15-01426-f008:**
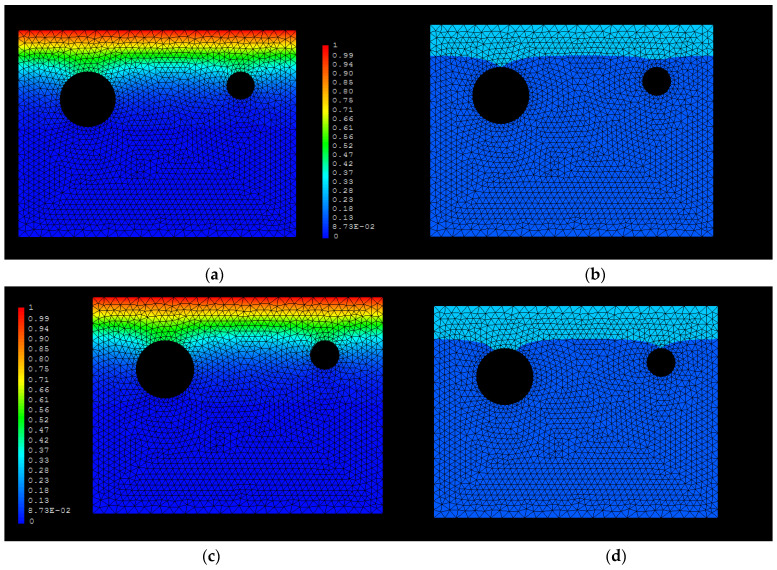
Diameter effect at the same concrete cover of 3 cm. (**a**) 10 years; (**b**) 10 years—threshold (0.4%); (**c**) 12 years; (**d**) 12 years—threshold (0.4%).

**Figure 9 materials-15-01426-f009:**
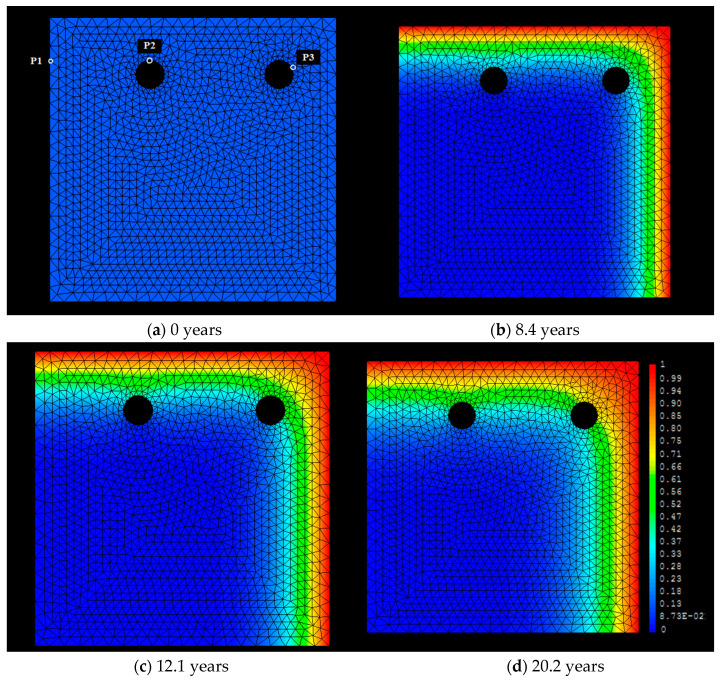
Penetration fronts of different chloride concentrations from two sides of the sample at different times. Three points, P1, P2 and P3, are specifically studied. The *C_x_* arrives first to the bar in the corner which accumulates more chlorides in its surface than the rest of bars.

**Figure 10 materials-15-01426-f010:**
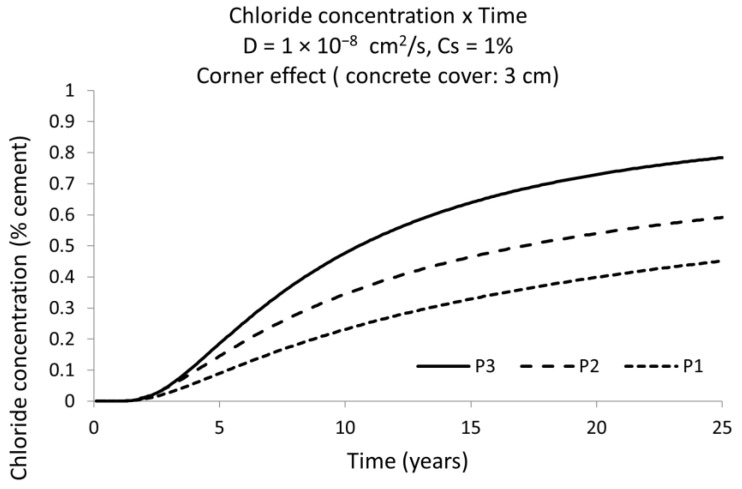
Chloride concentration versus time for the points P1, P2, and P3, shown in Figure 9 upper zone left side. P1 is placed in the bulk of concrete with no rebar. P2 is placed on the outmost exterior surface of the bar with unidirectional arrival of chlorides. P3 is on the outmost exterior surface of the bar but in the corner with bi-directional arrival of chlorides. The increase of chlorides with time is higher following P3 > P2 > P1.

**Figure 12 materials-15-01426-f012:**
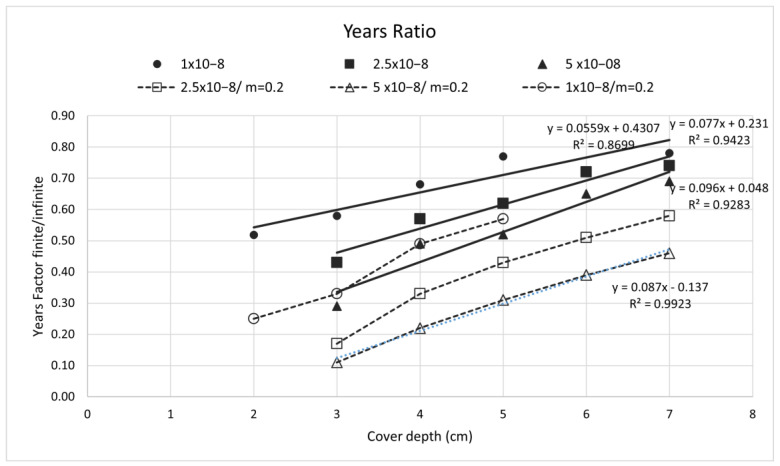
Ratio (in years of time to depassivation by a critical chloride content of 0.05% by concrete mass, in function of the cover depth for the diffusion coefficients of 1, 2.5 and 5 × 10^−8^ cm^2^/s.

**Figure 13 materials-15-01426-f013:**
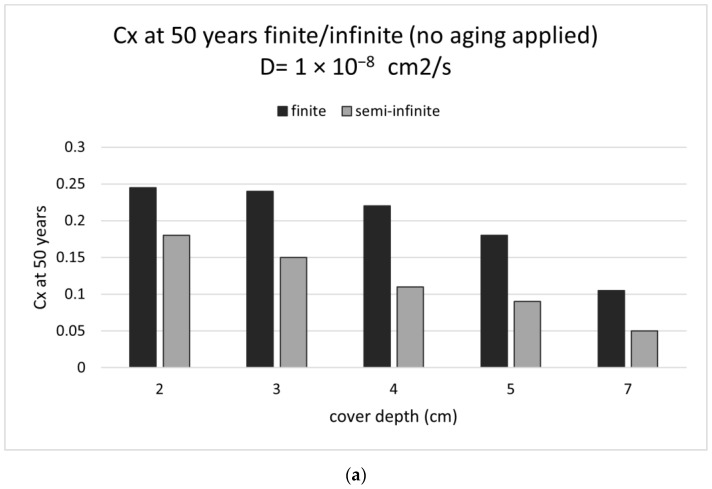
(**a**) Chloride contents at 50 years considering a D = 1 × 10^−8^ cm^2^/s and semi-infinite or finite conditions with respect to the cover depth (*x*-axis) for a surface concentration of 0.25% by weight of concrete and a bar diameter of 20 mm. The calculation was not considering an aging factor. (**b**) Ratio of finite/semi-infinite chloride content with respect to the cover depth for D = 1 and 2.5 × 10^−8^ cm^2^/s.

**Table 1 materials-15-01426-t001:** Service life time variation for different rebar diameters with the same concrete cover for the input parameters considered in the example.

Rebar Diameter	Time to Depassivation C_critical_ = 0.4%	Porcentual Increase in Chloride Concentration	Underestimation of Chloride Concentration
Erf. Function (semi-infinite boundary)	22.4 years	0%	100%
4 mm (finite boundary)	15.68 years	42.86%	30.00%
8 mm (finite boundary)	12.88 years	73.91%	42.50%
16 mm (finite boundary)	11.2 years	100.00%	50.00%
20 mm (finite boundary)	10.64 years	110.53%	52.50%

## Data Availability

Data is contained within the article.

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
