# Peer review of "Finite Boundary Conditions Due to the Bar Presence in the Model of Chloride Penetration"

_materials, 2022, doi:10.3390/ma15041426_

Round 1

Reviewer 1 Report

The research topic is interesting. A FEM model was applied to the chloride penetration. The propagation of the Steel-concrete interface was studied with the time, which could be helpful for the field of chloride-induced corrosion of the steel reinforcement in the concrete constructions. However, still some improvements are required in this manuscript. The reviewer thinks it can be considered if the authors can improve the manuscript carefully by considering the comments as follow:

  1. The presentation need improving. Almost all the instructions about the letters in the expressions are missed.
  2. The paper was presented over tedious, which reduces the quality of the global manuscript, including the conclusions.
  3. For the FEM model, more descriptions are required so as to improve the readable of the academic paper.
  4. Some further explanation and conclusions should be better involved by the simulation results. In this paper, the authors only presented the experimental results without enough analysis, such as Fig.2 and so on.

Author Response

They are in the attached file

Reviewer 2 Report

The topic of the research presented is novel and quite interesting to read. However, the technical writing of the paper need to be improved drastically.

  1. A thorough literature review needs to be discussed in the literature review section. Doesn't matter if they have not studied rebar location and size in their modelling. Literature on chloride concentration using Fick's law either experimentally and particularly modelling need to be shown to show the novelty of research.
  2. Eq 1 and 2- mention what each parameters represent.
  3. Page 2 line 34-40: What do you mean by skin? It is not common terminology, give proper context to understand what authors are trying to say.
  4. Experimental section lacks detailed methodology adopted for numerical simulation such as what are initial parameters assumed and where they are calculated from (experimentally or from literature)? Governing equations to define concrete and rebar properties etc.
  5. Page 2Line 59- Looks like authors are confused which barrier are they referring to, is it concrete surface or concrete-steel interface. In this para they first mentioned about concrete surface for diffusion using Galerkin weighted method and then they suddenly talked about barrier referring to interface. Please clarify in the manuscript.
  6. Since lack of information in methodology section, it is quite difficult to go through the results.
  7. Fig 2: if time is taken as constant of 25 years as mentioned on graph, why the x-axis shows time variation?
  8. Discussion needs to be presented in more detail. Also, compare it what is reported in past literature.
  9.  Typing error while citing the figure captions from fig 9 onwards.
  10. Quality of figures are poor. Figure 1, 11-13 are not clearly visible.
  11. Extensive English language corrections are required throughout the manuscript.
  12. Lack of references in the manuscript. Mainly because past literature chapters have not been referred.

Author Response

REVIEWER 1 Open Review

English language and style

( ) Extensive editing of English language and style required
(x) Moderate English changes required
( ) English language and style are fine/minor spell check required
( ) I don't feel qualified to judge about the English language and style

Yes

Can be improved

Must be improved

Not applicable

Does the introduction provide sufficient background and include all relevant references?

( )

(x)

( )

( )

Is the research design appropriate?

(x)

( )

( )

( )

Are the methods adequately described?

(x)

( )

( )

( )

Are the results clearly presented?

(x)

( )

( )

( )

Are the conclusions supported by the results?

( )

(x)

( )

( )

Comments and Suggestions for Authors

The research topic is interesting. A FEM model was applied to the chloride penetration. The propagation of the Steel-concrete interface was studied with the time, which could be helpful for the field of chloride-induced corrosion of the steel reinforcement in the concrete constructions. However, still some improvements are required in this manuscript. The reviewer thinks it can be considered if the authors can improve the manuscript carefully by considering the comments as follow:

  1. The presentation need improving. Almost all the instructions about the letters in the expressions are missed.

The only equations with no explanation of the symbols were eq. 1 and 2. We have added  the meaning of the symbols. Thanks for the reminder.

  1. The paper was presented over tedious, which reduces the quality of the global manuscript, including the conclusions.

There are tedious subjects. Numerical calculations are not very We do not know how to improve it.

  1. For the FEM model, more descriptions are required so as to improve the readable of the academic paper.

We are sorry but the referee does not say what specifically needs more information. We think that the calculations can be repeated by any person familiar with FEM and diffusion processes. Those not familiar with diffusion should not try to make the calculations.

  1. Some further explanation and conclusions should be better involved by the simulation results. In this paper, the authors only presented the experimental results without enough analysis, such as Fig.2 and so on.

The analysis has been made in the Introduction and in the different chapters. Figure 2 is self-explaining or we do not deduce how to explain it more than what we did. Repetition of the same arguments several times may be worse. Anyway, we have tried to introduce more detailed explanations and improve the language.

REVIEWER 2 open Review

English language and style

( ) Extensive editing of English language and style required
(x) Moderate English changes required
( ) English language and style are fine/minor spell check required
( ) I don't feel qualified to judge about the English language and style

Yes

Can be improved

Must be improved

Not applicable

Does the introduction provide sufficient background and include all relevant references?

( )

( )

(x)

( )

Is the research design appropriate?

( )

( )

(x)

( )

Are the methods adequately described?

( )

( )

(x)

( )

Are the results clearly presented?

( )

(x)

( )

( )

Are the conclusions supported by the results?

(x)

( )

( )

( )

Comments and Suggestions for Authors

The topic of the research presented is novel and quite interesting to read. However, the technical writing of the paper need to be improved drastically.

  1. A thorough literature review needs to be discussed in the literature review section. Sorry but we cannot agree because on one hand the “error function equation” is known from 40 years ago and extensively used in numerous research. Everybody in this subject knows the equation. On the other hand, there is a unique reference of a paper having a  similar approach to present one. This unique paper is referred here.  the Doesn't matter if they have not studied rebar location and size in their modelling. We cannot agree. It matters because we do not need to repeat unnecessarily the numerous papers on the typical solution to second Fick’s law but we need to emphasize the lack of studies (except the previous paper and us) on the facts presented in study.  Literature on chloride concentration using Fick's law either experimentally and particularly modelling need to be shown to show the novelty of research. The literature is very known for any person familiar with the chloride diffusion modelling. The only need is to mention the single reference with a similar approach to ours.
  2. Eq 1 and 2- mention what each parameters represent. Thanks, made.
  3. Page 2 line 34-40: What do you mean by skin? It is not common terminology, give proper context to understand what authors are trying to say. Again known the meaning in this area but we have explained more.
  4. Experimental section lacks detailed methodology adopted for numerical simulation such as what are initial parameters assumed and where they are calculated from (experimentally or from literature)? The values in a numerical solution are assumed. They can be found in experiments, but this paper is not experimental. We have assumed values currently observed. The cover depths are those of the standards. Governing equations to define concrete and rebar properties etc. They are not needed. The only needed is the differential equation and the boundary conditions which are mentioned several times along the text.
  5. Page 2Line 59- Looks like authors are confused which barrier are they referring to, is it concrete surface or concrete-steel interface. There is not a confusion from our part. The bar acts as a barrier for chlorides diffusing inside the concrete. In this para they first mentioned about concrete surface for diffusion using Galerkin weighted method and then they suddenly talked about barrier referring to interface. We are sorry the referee does not understand but the explanation is correct. Please clarify in the manuscript.
  6. Since lack of information in methodology section, it is quite difficult to go through the results. We are afraid that perhaps the referee is not knowing chloride diffusion calculations.
  7. Fig 2: if time is taken as constant of 25 years as mentioned on graph, why the x-axis shows time variation? Again we are afraid that perhaps the referee is not knowing chloride diffusion calculations
  8. Discussion needs to be presented in more detail. Also, compare it what is reported in past literature. There is not more other literature than a paper with similar conclusions. We have tried to improve the language and extend some explanations
  9.  Typing error while citing the figure captions from fig 9 onwards. We have corrected.
  10. Quality of figures are poor. Figure 1, 11-13 are not clearly visible. Figure 1 is what is given by the FEM program. We have improved figure 11 by giving another version.
  11. Extensive English language corrections are required throughout the manuscript. We have tried to improve but without specifying what is wrong it is very difficult where to concentrate the improvement
  12. Lack of references in the manuscript. Mainly because past literature chapters have not been referred. It is not needed too many references as said before. The equations are in many places and books

We thank the referees for the time taken to read the paper but unfortunately their comments did not help too much.

Reviewer 3 Report

  1. In analysing the problem to be dealt with in the research work (Introduction), each of the referenced sources should be accompanied by a critical comment (at least one sentence). The manuscript contains old sources of information (older than ten years), thus making a purpose of the research under doubt. The reference 17 is incomplete.
  2. The chloride flow J(x) is determined under the first Fick's law and it can not be treated as similar with the rate of changes of chlorides concentration in the equation (1), which is described by the second Fick's law.
  3. The curves given in Fig.2 have no designation and do not reflect fully the obtained and described result.
  4. The designation of the curves in Fig. 4 is wrong.
  5. In Figures 4 and 5, the "error function" curve is not labeled.
  6. Unit of measurement in Fig.10 should be „years”.
  7. Images in Fig. 11 are not readable and for this reason bear no useful information.
  8. Data contained in Table 1 are difficult to analyse, because the columns have identical headings.

9. There is no analysis of the results given in Fig.12.

Author Response

We thank very much the referee for the valuable indications

Reviewer 4 Report

In this study, the effect of physical presence of reinforcing bars on salt penetration into reinforced concrete members was analytically examined. Though it is regrettable that there is no comparison with the actual salt penetration situation, the interesting examination result has been obtained. In addition, it would have been better if the analysis had been carried out considering that the structure of the concrete at the reinforcing bar interface is different from the bulk part.

The following editorial minor comments should be addressed.

Line 9: "second Fick's law" should be "Fick's second law"

Line 106: "variacion" shoud be "variation" ??

Line 103: The definitions of x and t should be given here.

equation 2: The left side of the equation should be C, not Cth.

Line 116: Delete this line, and add the following sentence into Line 120: "When C reaches Cth, the steel is depassivated."

Line 122: "side" should be "direction"

Figure 2: "Cs=1" should be "Cs=1%"

Figures 3, 4, 5, and 10: "Cs=1%" should be added in these figures.

Line 225: "os" should be "of"??

Lines 226, 229, 244 and 247: link errors!! Did you check your manuscript enough before submit it??

Figure 10: The label for y axis lacks unit (% cement)

Line 270: "n=4"  1) It should be "m=4". 2)If you mention the aging factor here, the definition for it in Lines 289-294 should be moved to the end of Section 3.

Line 275: "factor" I think "ratio" is more appropriate.

Line 340 and 341: There are two Reference 1.

Author Response

We wonder if we have understood well the first paragraph of the reviewer because it seems to us that our work has not been understood. We goo sentence by sentence. We thank the reviewer for the identification of errors. In our manuscript there are not numbers in the lines.

….IN THIS STUDY, THE EFFECT OF PHYSICAL PRESENCE OF REINFORCING BARS ON SALT PENETRATION INTO REINFORCED CONCRETE MEMBERS WAS ANALYTICALLY EXAMINED….we analyzed numerically and not analytically, perhaps “analytically” from the reviewer has a grammatical meaning and not a mathematical meaning but we would like to stress that our program is numerical

……THOUGH IT IS REGRETTABLE THAT THERE IS NO COMPARISON WITH THE ACTUAL SALT PENETRATION SITUATION, THE INTERESTING EXAMINATION RESULT HAS BEEN OBTAINED…..We have made that comparison in each of the figures 3 to 5 and 6. What is named as “salt penetration by the referee” is what we name in the figure “error function equation. The comparison is the basis of our work.

…..IN ADDITION, IT WOULD HAVE BEEN BETTER IF THE ANALYSIS HAD BEEN CARRIED OUT CONSIDERING THAT THE STRUCTURE OF THE CONCRETE AT THE REINFORCING BAR INTERFACE IS DIFFERENT FROM THE BULK PART….. Just is what we did and our program does. Figures 8 to 11 just show the effect in the steel interface with respect to the bulk of the concrete.

LINE 9: "SECOND FICK'S LAW" SHOULD BE "FICK'S SECOND LAW" corrected

LINE 106: "VARIACION" SHOUD BE "VARIATION" ?? corrected

LINE 103: THE DEFINITIONS OF X AND T SHOULD BE GIVEN HERE. corrected

EQUATION 2: THE LEFT SIDE OF THE EQUATION SHOULD BE C, NOT CTH- changed by Cx corrected in all the text.

LINE 116: DELETE THIS LINE, AND ADD THE FOLLOWING SENTENCE INTO LINE 120: "WHEN C REACHES CTH, THE STEEL IS DEPASSIVATED." made

LINE 122: "SIDE" SHOULD BE "DIRECTION" made

FIGURE 2: "CS=1" SHOULD BE "CS=1%" corrected

FIGURES 3, 4, 5, AND 10: "CS=1%" SHOULD BE ADDED IN THESE FIGURES.- made

LINE 225: "OS" SHOULD BE "OF"?? made

LINES 226, 229, 244 AND 247: LINK ERRORS!! DID YOU CHECK YOUR MANUSCRIPT ENOUGH BEFORE SUBMIT IT?? We do not know why the “link errors” appeared. They are not in our manuscript: that which we submitted. In this line we have added that the external concentration is equal in the two sides can be also seen in figure 9c and d and that also sows the internal concentration near the bars.

FIGURE 10: THE LABEL FOR Y AXIS LACKS UNIT (% CEMENT)- made

LINE 270: "N=4"  1) IT SHOULD BE "M=4".made

 2)IF YOU MENTION THE AGING FACTOR HERE, THE DEFINITION FOR IT IN LINES 289-294 SHOULD BE MOVED TO THE END OF SECTION 3.-made

LINE 275: "FACTOR" I THINK "RATIO" IS MORE APPROPRIATE. -made and changed in the vertical axis of figure 13

LINE 340 AND 341: THERE ARE TWO REFERENCE 1.corrected

Round 2

Reviewer 1 Report

The authors could not well respond to the comments as provided in the first review.

Author Response

(The authors gave the same response as above.)

Reviewer 2 Report

Instead of properly working on the suggested comments. Authors are trying to defend themselves by rudely answering to reviewer queries. Very little improvement has been made from the previous version. Response to reviewer queries has not been properly answered.

Author Response

(The authors gave the same response as above.)
